# Factors associated with low-level viraemia in people with HIV starting antiretroviral therapy: A Swedish observational study

**Hanna Brattgård**[1], **Per Björkman**[1,2], **Piotr Nowak**[3,4], **Carl Johan Treutiger**[5], **Magnus Gisslén**[6,7], **Olof Elvstam**[1,8]*

1 Department of Translational Medicine, Lund University, Malmö, Sweden, 2 Department of Infectious Diseases, Skåne University Hospital, Malmö, Sweden, 3 Department of Infectious Diseases, Karolinska University Hospital, Stockholm, Sweden, 4 Department of Medicine Huddinge, Unit of Infectious Diseases, Karolinska Institutet, Stockholm, Sweden, 5 Department of Infectious Diseases/Venhälsan, South General Hospital, Stockholm, Sweden, 6 Department of Infectious Diseases, Institute of Biomedicine, Sahlgrenska Academy, University of Gothenburg, Gothenburg, Sweden, 7 Department of Infectious Diseases, Region Västra Götaland, Sahlgrenska University Hospital, Gothenburg, Sweden, 8 Department of Infectious Diseases, Växjö Central Hospital, Växjö, Sweden

* olof.elvstam@med.lu.se

**Data Availability Statement:** Data cannot be shared publicly. Data are available from the quality register InfCareHIV for researchers who meet the criteria for access to confidential data. We have

## Abstract

### Objective

Low-level viraemia (LLV) occurs in some people with HIV (PWH) receiving antiretroviral therapy (ART) and has been linked to inferior treatment outcomes. We investigated factors associated with LLV in a nationwide cohort of Swedish PWH starting ART.

### Methods

Participants were identified from the InfCareHIV register, with the following inclusion criteria: ART initiation 2006–2017, age >15 years, ≥4 viral load (VL) results available and no documented treatment interruptions or virologic failure (≥2 consecutive VL ≥200 copies/ml) during follow-up. Starting from 6 months after ART initiation, participants were followed for 24 months and categorised as viral suppression (VS; VL <50 copies/ml) or LLV (≥2 consecutive VL 50–199 copies/ml). We analysed the association between the following factors and LLV using multivariable logistic regression: sex, age, pre-ART VL and CD4 count, ART regimen, country of birth, HIV-1 subtype and transmission category.

### Results

Among 3383 participants, 3132 (92.6%) had VS and 251 (7.4%) had LLV. In univariable analyses, factors associated with LLV were male sex, higher age, lower pre-ART CD4 count, higher pre-ART VL and ART regimen. After adjustment, the following factors were associated with LLV (adjusted odds ratio; 95% confidence interval): male sex (1.6; 1.1–2.3), higher pre-ART VL (2.7; 2.2–3.3), pre-ART CD4 count <200 cells/μl (1.6; 1.2–2.2), protease inhibitor (PI)-based regimen (1.5; 1.1–2.1), non-standard ART (2.4; 1.0–5.5) and injecting drug use (2.0; 1.1–3.7).

prepared a dataset underlying the findings in this manuscript and deposited this file within the steering committee of InfCareHIV (contact via Dr. Christina Carlander, Department of Infectious Diseases, Karolinska University Hospital, Huddinge, SE-14186, Stockholm, Sweden, email: christina.carlander@regionstockholm@se).

**Funding:** This study was supported by a grant (0825-011 8298; to OE) from Department of Research and Development, Region Kronoberg, Växjö. The funders had no role in study design, data colletion and analysis, decision to publish, or preparation of the manuscript.

**Competing interests:** I have read the journal's policy and the authors of this manuscript have the following competing interests: PB has received grants from Swedish State, grants from Region Skåne, grants from Gilead Nordic Fellowship during the conduct of the study, and personal fees from Gilead, outside the submitted work. CT has received honoraria as speaker and/or advisor from GlaxoSmithKline/ViiV, outside the submitted work. MG has received research grants from Gilead Sciences and Janssen-Cilag and honoraria as speaker and/or scientific advisor from Amgen, Biogen, Bristol-Myers Squibb, Gilead Sciences, GlaxoSmithKline/ViiV, Janssen-Cilag, MSD, Novocure, Novo Nordic and Sanofi, outside the submitted work. OE has received grant from Pfizer, outside the submitted work. The manuscript does not include patents, products in development or marketed products, and none of the authors report employment or consultancy as competing interest for this work. This does not alter our adherence to PLOS ONE policies on sharing data and materials.

## Conclusion

Among Swedish PWH, LLV during ART was associated with markers of HIV disease severity before starting ART, male sex, injecting drug use and use of PI-based or non-standard ART regimens.

## Introduction

Although most people with HIV (PWH) receiving antiretroviral therapy (ART) achieve persistent viral suppression, some individuals have low but detectable plasma levels of HIV RNA without meeting criteria for virologic failure, a phenomenon commonly referred to as low-level viraemia (LLV) [1]. LLV could arise from two separate mechanisms: either passive release of viral particles from latently infected cells or as a consequence of ongoing replication [2–4]. LLV confers increased risk of virologic failure [5–9] and has also been associated with all-cause mortality in a recent analysis of the nationwide Swedish HIV cohort [10]. In order to interpret these associations, it is important to understand the factors that may influence development of LLV during ART.

Parameters reflecting HIV disease severity at ART initiation (high viral load [VL] and low CD4 count) have been associated with LLV in some studies [11–13], in which associations with sex and HIV acquisition route were also reported [12, 13]. Apart from host and viral characteristics, treatment-related factors might influence the risk of LLV, with higher risk for protease inhibitor (PI)-based regimens compared with those based on non-nucleoside reverse transcriptase inhibitors (NNRTIs) or integrase strand transfer inhibitors (INSTIs) [11, 14, 15]. However, these studies were conducted before introduction of INSTI-based regimens for first-line ART, and the potential association between ART regimen and risk of LLV thus remains uncertain for contemporary treatment practices.

Here, we aimed to investigate viral, host and treatment-related factors with regard to development of LLV in a nationwide cohort of Swedish PWH followed after ART initiation.

## Materials and methods

### Patient selection

We used data from the nationwide InfCareHIV register, a quality register encompassing >99% of Swedish PWH [16]. This register contains information on demographics, HIV acquisition route, CD4 counts, VL, HIV-1 subtype, antiretroviral drugs prescribed and results from genotypic resistance testing [17]. We included participants who started ART ($\geq$3 non-booster drugs, of which not all were nucleoside/nucleotide reverse transcriptase inhibitors) January 1, 2006, to December 31, 2017, were >15 years old at ART initiation and had $\geq$ 4 VL measurements during the period 6–30 months after initiation of ART. Participants with documented treatment interruptions or virologic failure (defined as $\geq$2 consecutive VL measurements $\geq$200 copies/ml or any VL $\geq$1,000 copies/ml) during the 24 months observation period were excluded, as well as individuals with HIV-2 infection. The Regional Ethics Committee of Lund, Sweden, approved the study (2017/1023). No specific consent was deemed required for this study; data were pseudo-anonymized.

### Definition of viraemia categories

Participants were divided into two categories based on their longitudinal viraemia profile during the observation period: viral suppression (VS) and LLV. VS was defined as VL <50 copies/ml and LLV was defined as ≥2 consecutive VL measurements of 50–199 copies/ml with >4 weeks interval (or one VL in the range 50–199 and one in the range 50–999 copies/ml, thus not meeting the definition of virologic failure). Persons with isolated VL measurements of 50–999 copies/ml preceded and followed by VL<50 copies/ml were included in the VS category.

### Statistical analysis

The following variables were considered: sex, age at ART initiation, pre-ART CD4 count (higher/lower than 200 cells/μl) and VL (modelled logarithmically), route of transmission (heterosexual contact, male-to-male sexual contact, injecting drug use and other), country of birth (Nordic countries [Sweden, Denmark, Norway, Finland and Iceland] yes/no), HIV subtype (B/non-B) and type of initial ART regimen (NNRTI-based, PI-based, INSTI-based and non-standard ART [including regimens with more than one anchor drug or fusion/entry inhibitors]). Separately, we also considered CD4 percentage, rather than absolute count, and the CD4:CD8 ratio.

Data were analysed using binary logistic regression. Variables included in the main multivariable model were sex, age, country of birth, transmission category, pre-ART VL, pre-ART CD4 count and ART regimen. The association between HIV subtype and LLV was analysed separately, with adjustment for country of birth and transmission route, in addition to sex and age. For the multivariable analyses, correlation matrices were made with the included independent variables to check for multicollinearity. To investigate if the same factors were associated with LLV in men and women with HIV, respectively, we separately performed an analysis stratified by sex. Likewise, since the management of HIV has changed considerably during the studied years, we performed an analysis stratified by year of treatment initiation (before/after January 1, 2014, the year when dolutegravir became available in Sweden). Missing data were handled using complete case analysis, and we report the number of missing values per each variable. As a sensitivity analysis, age was also modelled using restricted cubic splines. We used IBM SPSS 27 Statistics for Windows, version 10 (IBM Corp., Armonk, NY, United States) and Stata SE/15.1 for Windows (StataCorp LLC, College Station, TX, United States) and report all odds ratios (OR) with 95% confidence intervals (CI). Fig 2 was constructed using R [18] with the ggplot2 package [19].

## Results

### Participant characteristics

A total of 5821 persons started ART between January 1, 2006, and December 31, 2017. Of these, 2438 (41.9%) were excluded (Fig 1). Of 251 individuals excluded since they met the definition of virologic failure during the observation period, 16 (6.4%) had LLV before reaching failure. Among the remaining 3383 participants, 3132 (92.6%) had VS, and 251 (7.4%) met the study definition of LLV. Approximately similar proportions of LLV were observed across calendar year of ART initiation (Fig 2). The median age at ART initiation was 38 years and 62.6% of the study participants were male. The most common transmission route was heterosexual contact (52.7%), and acquisition through injecting drugs was reported in 141 cases (4.2%). A majority (62.7%) of the participants had pre-ART CD4 count >200 cells/μl, and the median pre-ART VL was 4.8 $\log_{10}$ copies/ml (Table 1). The median number of VL measurements

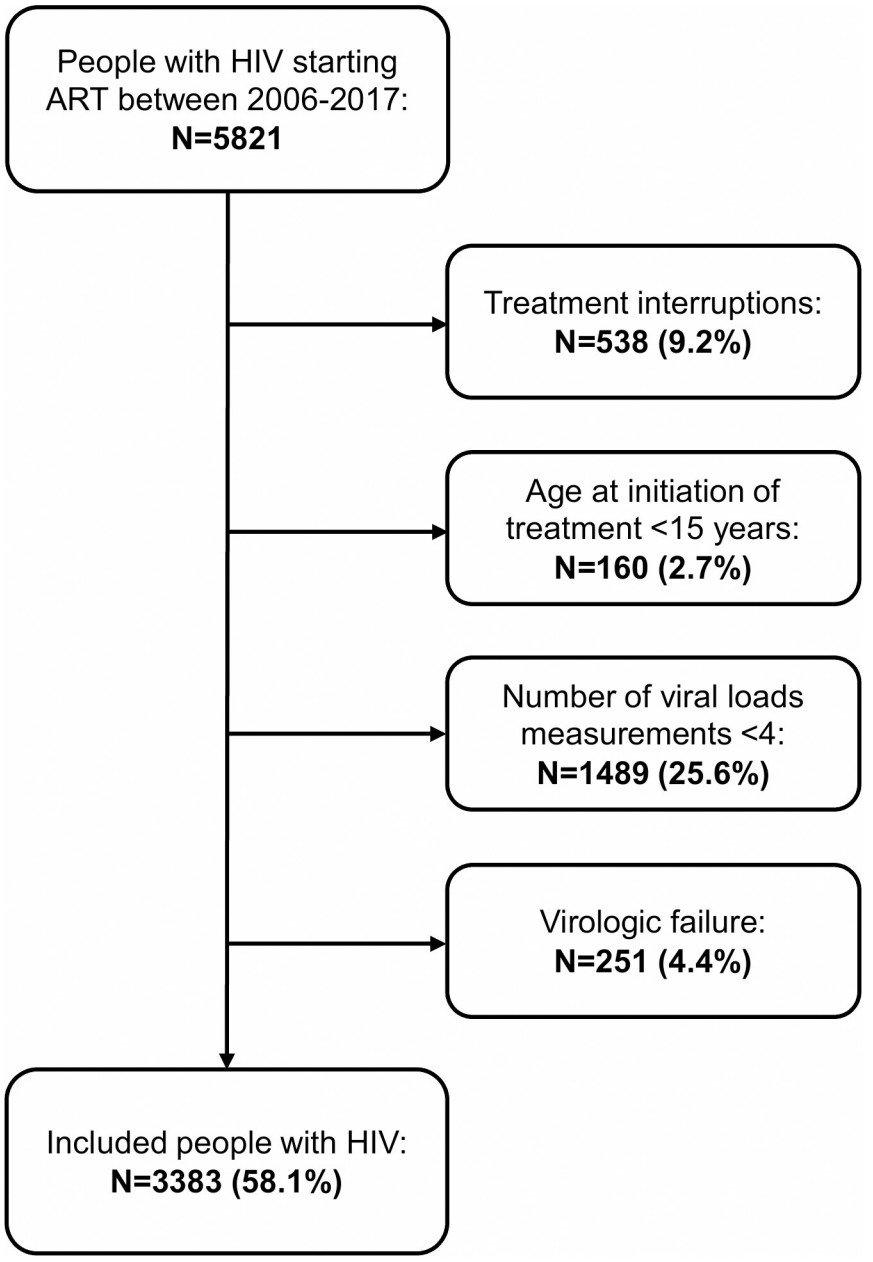

**Fig 1. Flow chart of study inclusion and exclusion.** Abbreviation: ART, antiretroviral therapy.

during the study period after starting ART was 5 for VS and 7 for LLV. The median VL during follow-up was <20 copies/ml for VS and 46 copies/ml for LLV.

## Factors associated with LLV

The following factors were associated with LLV in univariable analyses: sex, age, pre-ART CD4 count, pre-ART VL and PI-based and non-standard regimen (compared with NNRTI-based). When adjusting for sex, age, pre-ART CD4 count and VL, transmission category, country of birth and ART regimen, a statistically significant association to LLV was found for

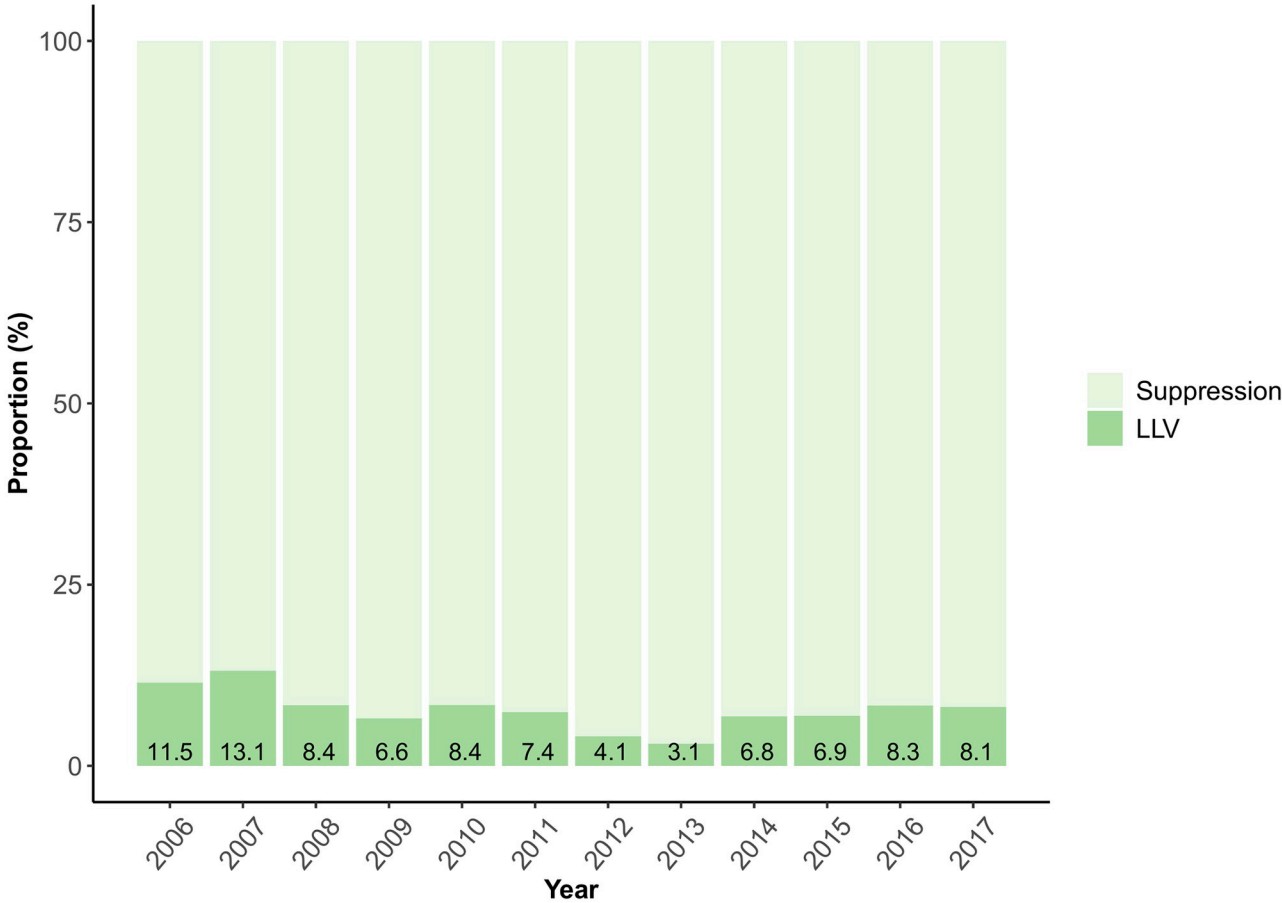

**Fig 2. Proportion of study participants experiencing LLV during 6–30 months after start of ART, grouped by year of ART initiation.**
Abbreviations: ART, antiretroviral therapy; LLV, low-level viraemia.

male sex (adjusted OR [aOR], 1.6; 95% CI, 1.1–2.3), pre-ART CD4 count <200 cells/µl (aOR, 1.6; 95% CI, 1.2–2.2), high pre-ART VL (aOR, 2.7; 95% CI, 2.2–3.3), PI-based regimen (aOR, 1.5; 95% CI, 1.1–2.1), non-standard ART (aOR, 2.4; 95% CI, 1.0–5.5) and injecting drug use (aOR, 2.0; 95% CI, 1.1–3.7) (Table 2). Modelling age with restricted cubic splines did not change these results.

Replacing absolute CD4 counts with CD4 percentages did not substantially change our conclusions regarding the other variables; having higher CD4 percentage was linked to lower odds of LLV (aOR, 0.98; 95% CI, 0.96–0.99). Having a low CD4:CD8 ratio was associated with higher odds of LLV in univariable analysis, but when adjusted for absolute CD4 counts and the other variables in the full model, this was not statistically significant (aOR, 0.9; 95% CI, 0.6–1.4).

In a subanalysis restricted to participants with available HIV subtype, we observed no statistically significant association between HIV subtype B (versus non-B) and the risk of LLV, neither in univariable analysis or after adjustment for sex, age, country of birth and suspected route of transmission (aOR, 0.8; 95% CI, 0.5–1.3).

Furthermore, in a subanalysis stratified by sex, the association between pre-ART CD4 count and LLV was not observed for females (Table 3). The association between higher pre-ART VL and higher odds of developing LLV was statistically significant in both men and

**Table 1. Characteristics of study participants.**

| | Total n = 3383 | Virological suppression n = 3132 (92.6%) | Low-level viraemia n = 251 (7.4%) |
|---|---|---|---|
| Male | 2119 (62.6%) | 1941 (62.0%) | 178 (70.9%) |
| Age at ART initiation (years) | 38 (32–47) | 38 (31–47) | 40 (33–48) |
| ART initiation >2014 | 1016 (30.0%) | 940 (30.0%) | 76 (30.3%) |
| Country of birth | | | |
| Sweden | 1176 (34.8%) | 1083 (34.6%) | 93 (37.1%) |
| Nordic countries | 1255 (37.1%) | 1158 (37.0%) | 97 (38.7%) |
| Non-Nordic countries | 2093 (61.9%) | 1940 (61.9%) | 153 (61.0%) |
| Missing | 35 (1.0%) | 34 (1.1%) | 1 (0.4%) |
| Transmission category | | | |
| Heterosexual contact | 1783 (52.7%) | 1653 (52.8%) | 130 (51.8%) |
| Male-to-male sexual contact | 1115 (33.0%) | 1038 (33.1%) | 77 (30.7%) |
| Injecting drug use | 141 (4.2%) | 126 (4.0%) | 15 (6.0%) |
| Mother-child | 16 (0.5%) | 16 (0.5%) | 0 |
| Blood products | 34 (1.0%) | 34 (1.1%) | 0 |
| Other/Unknown | 232 (6.9%) | 206 (6.6%) | 26 (10.4%) |
| Missing | 62 (1.8%) | 59 (1.9%) | 3 (1.2%) |
| HIV-specific characteristics | | | |
| Pre-ART CD4 count (cells/μl) | 270 (150–390) | 273 (160–400) | 162 (67–260) |
| Pre-ART CD4 count <200 cells/μl | 1070 (31.6%) | 932 (29.8%) | 138 (55.0%) |
| Missing | 193 (5.7%) | 181 (5.8%) | 12 (4.8%) |
| Pre-ART CD4 percentage (%) | 23 (16–30) | 23 (16–30) | 18 (12–25) |
| Missing | 182 (5.4%) | 168 (5.4%) | 14 (5.6%) |
| Pre-ART CD4:CD8 ratio | 0.45 (0.28–0.70) | 0.46 (0.29–0.70) | 0.35 (0.20–0.55) |
| Missing | 608 (18.0%) | 556 (17.8%) | 52 (20.7%) |
| Pre-ART VL ($\log_{10}$ copies/ml) | 4.8 (4.2–5.4) | 4.7 (4.1–5.3) | 5.5 (5.1–6.0) |
| Pre-ART VL ≥100 000 copies/ml | 1255 (37.1%) | 1069 (34.1%) | 186 (74.1%) |
| Missing | 210 (6.2%) | 199 (6.4%) | 11 (4.4%) |
| HIV-1 subtype | | | |
| A | 54 (1.6%) | 50 (1.6%) | 4 (1.6%) |
| B | 801 (23.7%) | 750 (24.0%) | 51 (20.3%) |
| C | 349 (10.3%) | 324 (10.3%) | 25 (10.0%) |
| CRF | 538 (15.9%) | 495 (15.8%) | 43 (17.1%) |
| Other | 136 (4.0%) | 120 (3.8%) | 16 (6.4%) |
| Missing | 1505 (44.5%) | 1393 (44.5%) | 112 (44.6%) |
| ART regimen | | | |
| NNRTI-based | 1431 (42.3%) | 1351 (43.1%) | 80 (31.9%) |
| PI-based | 1261 (37.3%) | 1143 (36.5%) | 118 (47.0%) |
| INSTI-based | 634 (18.7%) | 590 (18.8%) | 44 (17.5%) |
| Non-standard[a] | 57 (1.7%) | 48 (1.5%) | 9 (3.6%) |

Abbreviations: ART, antiretroviral therapy; CRF, circulation recombinant forms; INSTI, integrase strand transfer inhibitor; NNRTI, non-nucleoside reverse transcriptase inhibitor; PI, protease inhibitor; VL, viral load.

[a] Including regimens with more than one anchor drug (n = 54) or regimens based on the fusion inhibitor enfuvirtide (n = 1) or the CCR5 antagonist maraviroc (n = 2).

Values are No. (%) or median (interquartile range).

**Table 2. Factors associated with low-level viraemia in univariable and multivariable analyses.**

| | Unadjusted OR | Adjusted OR[a] |
|---|---|---|
| Male sex | 1.5 (1.1–2.0) | 1.6 (1.1–2.3) |
| Age at ART initiation | 1.02 (1.00–1.03) | 1.00 (0.99–1.02) |
| Pre-ART CD4 <200 cells/μl | 3.0 (2.3–3.9) | 1.6 (1.2–2.2) |
| Pre-ART VL (modelled logarithmically) | 3.0 (2.5–3.6) | 2.7 (2.2–3.3) |
| ART regimen | | |
| NNRTI-based (Ref) | | |
| PI-based | 1.7 (1.3–2.3) | 1.5 (1.1–2.1) |
| INSTI-based | 1.3 (0.9–1.8) | 1.0 (0.7–1.5) |
| Non-standard[b] | 3.2 (1.5–6.7) | 2.4 (1.0–5.5) |
| Transmission category | | |
| Male-to-male sexual contact (Ref) | | |
| Heterosexual contact | 1.1 (0.8–1.4) | 1.3 (0.9–1.9) |
| Injecting drug use | 1.6 (0.9–2.9) | 2.0 (1.1–3.7) |
| Other | 1.4 (0.9–2.2) | 1.7 (1.0–3.0) |
| Born outside the Nordic countries | 0.9 (0.7–1.2) | 1.2 (0.8–1.7) |

Abbreviations: ART, antiretroviral therapy; INSTI, integrase strand transfer inhibitor; NNRTI, non-nucleoside reverse transcriptase inhibitor; OR, odds ratio; Ref, reference; PI, protease inhibitor; VL, viral load.

[a] 3003 cases included in the multivariable analysis. Adjusted for the variables in the table.

[b] Including regimens with more than one anchor drug or regimens based on fusion/entry inhibitors.

**Table 3. Association between low-level viraemia and pre-ART CD4 count, pre-ART viral load and type of ART, stratified by sex.**

| | Females: n = 1109 Adjusted OR[a] | Males: n = 1933 Adjusted OR[a] |
|---|---|---|
| Age at ART initiation | 1.00 (0.98–1.03) | 1.01 (0.99–1.02) |
| Pre-ART CD4 <200 cells/μl | 1.1 (0.6–1.9) | 2.1 (1.4–2.9) |
| Pre-ART VL (modelled logarithmically) | 2.2 (1.5–3.1) | 3.1 (2.5–3.9) |
| ART regimen | | |
| NNRTI-based (Ref) | | |
| PI-based | 2.2 (1.2–4.2) | 1.2 (0.8–1.8) |
| INSTI-based | 1.0 (0.4–2.5) | 0.9 (0.6–1.5) |
| Non-standard[b] | 5.0 (1.2–20.6) | 1.5 (0.5–4.4) |
| Born outside the Nordic countries | 1.0 (0.4–2.0) | 1.3 (0.9–1.9) |

Abbreviations: ART, antiretroviral therapy; INSTI, integrase strand transfer inhibitor; NNRTI, non-nucleoside reverse transcriptase inhibitor; OR, odds ratio; Ref, reference; PI, protease inhibitor; VL, viral load.

[a] Adjusted for the variables in the table.

[b] Including regimens with more than one anchor drug or regimens based on fusion/entry inhibitors.

women, and in both sexes starting ART before and after 2014. The increased odds of LLV in injecting drug users was restricted to persons initiating ART before 2014 (Table 4).

## Discussion

In this study, based on a nationwide cohort of Swedish PWH, LLV during ART was associated with higher pre-ART VL and lower CD4 count, as well as with male sex, injecting drug use and use of PI-based or non-standard ART regimens.

**Table 4. Association between low-level viraemia and pre-ART CD4 count, pre-ART viral load and type of ART, stratified by starting ART before/after 2014.**

|  | Before 2014: n = 2156 Adjusted OR[a] | After 2014: n = 847 Adjusted OR[a] |
|---|---|---|
| Male sex | 1.2 (0.8–1.8) | 3.0 (1.5–6.2) |
| Age at ART initiation | 1.01 (0.99–1.02) | 1.00 (0.98–1.03) |
| Pre-ART CD4 <200 cells/µl | 1.5 (1.0–2.1) | 1.9 (1.1–3.3) |
| Pre-ART VL (modelled logarithmically) | 3.0 (2.3–3.8) | 2.3 (1.7–3.2) |
| ART regimen |  |  |
| NNRTI-based (Ref) |  |  |
| PI-based | 1.4 (1.0–2.1) | 1.6 (0.7–3.6) |
| INSTI-based | 0.5 (0.1–2.1) | 0.8 (0.4–1.6) |
| Non-standard[b] | 3.1 (1.2–7.8) | 0.8 (0.1–7.6) |
| Transmission category |  |  |
| Male-to-male sexual contact (Ref) |  |  |
| Heterosexual contact | 1.7 (1.0–2.6) | 0.8 (0.4–1.6) |
| Injecting drug use | 2.3 (1.1–4.6) | 1.2 (0.2–6.7) |
| Other | 1.9 (0.9–3.8) | 1.4 (0.6–3.4) |
| Born outside the Nordic countries | 1.0 (0.6–1.5) | 1.6 (0.8–3.1) |

Abbreviations: ART, antiretroviral therapy; INSTI, integrase strand transfer inhibitor; NNRTI, non-nucleoside reverse transcriptase inhibitor; OR, odds ratio; Ref, reference; PI, protease inhibitor; VL, viral load.

[a] Adjusted for the variables in the table

[b] Including regimens with more than one anchor drug or regimens based on fusion/entry inhibitors.

 LLV, defined as VL of 50–199 copies/ml, occurred in 7.4% of participants, illustrating that this phenomenon is relatively common in persons receiving modern ART regimens. Although differences in the definition of LLV, cohort composition and follow-up time preclude direct comparisons between studies, the frequency of LLV in our cohort was comparable to contemporary reports from other high-income settings [11–13].

 Our findings of associations between higher pre-ART VL and lower pre-ART CD4 (both absolute counts and percentage) and LLV are in agreement with previous studies [11–13, 15]. Both these factors reflect more advanced HIV infection, which suggests that LLV may be related to a larger HIV reservoir [20], and that passive release of virions from latently infected cells could be an important mechanism for LLV in our study population [2, 3]. Apart from these HIV-specific factors, we also observed that LLV was more common in men than in women. This finding is in line with studies conducted in Spain and Austria [12, 13]. In the Austrian study, associations with sex were stratified by transmission route, which showed that females infected through heterosexual contact had lower risk of LLV compared with men who have sex with men [13]. In line with this previous study [13], we observed higher risk of LLV in people with injecting drug use as their transmission category. However, this association was not statistically significant in the subanalysis of people starting ART after 2014. This might reflect changes in the Swedish HIV epidemic and/or better management of HIV in people who inject drugs during the study period. However, the number of people infected with HIV through injecting drug use was small (4.2% of the study population), so the findings of this subanalysis should be interpreted with caution. In our population, the association between sex and LLV was restricted to persons starting ART after 2014, although the proportions of men and women were similar during the whole study period. It is possible that the association

between LLV and sex reflects differences in disease severity at HIV diagnosis between men and women, for instance related to late presentation [21].

Previous studies have found higher risk of LLV among people receiving PI-based compared to NNRTI-based ART [11, 12, 14, 15, 22]. We also observed such an association, but in the subanalyses restricted to participants starting ART before or after 2014, this relationship was not statistically significant. The association between different regimens and viraemia profile during ART might be influenced by indication bias, with higher likelihood of PI prescription in persons with anticipated lower adherence [11, 13]. Yet, the association between PI and LLV has also been observed in a retrospective analysis of two randomized trials [23] and is also supported by increased occurrence of viral blips in PWH on PI-based compared to other regimens [24]. We also observed higher odds of LLV among people with non-standard ART regimens (including combinations of more than one core agent and fusion or entry inhibitors). The number of patients with these regimens was small, and we consider it likely that they reflect a special category. For example, detection of pre-ART drug resistance mutations may have led to use of non-standard initial regimen. Some of these persons could also have undocumented previous ART experience. Bernal et al. reported lower frequency of LLV in persons receiving INSTI-based regimens [12], but in our material, we found no indication that INSTI-based ART confers reduced risk of LLV compared with other regimens.

Since 2015, ART is recommended for all PWH, irrespective of disease severity [25, 26]. In Swedish guidelines, ART has been recommended irrespective of CD4 counts since 2014 [27]. Since high pre-ART VL and low CD4 count were major risk factors for LLV in our material, earlier ART initiation could be expected to result in reduced incidence of LLV. Still, advanced disease at HIV diagnosis remains common in Europe [28], so improved testing strategies are likely needed to consolidate the effect of early ART on the occurrence of LLV on a population level.

The main limitation of this study concerns residual confounding. We adjusted for several potential confounders, but the database did not contain information on some factors which have been associated with LLV in other studies, such as socioeconomic status or treatment adherence [29]. Furthermore, misclassification of viraemia category might have occurred due to relatively infrequent sampling (reflecting clinical practice). For example, it is possible that some individuals categorized as virally suppressed may have had unrecognized shorter periods of LLV. Also, some individuals with LLV were excluded since they developed failure; our results are therefore restricted to people with LLV without treatment failure during the first years after ART initiation. Lastly, while previous data indicate that type of VL assay could be associated with LLV [13], we lack information on which assays that were used for specific VL measurements. The major strength of our study, which is the first to describe factors associated with LLV in Northern Europe, is that it is based on a nationwide cohort of PWH. Furthermore, we present characteristics of LLV among patients starting ART during a period with access to modern INSTI-based regimens.

## Conclusion

Among Swedish PWH initiating ART between 2006 and 2017, markers of HIV disease severity at ART initiation were associated with higher risk of LLV. This risk was also increased in men, people who acquired HIV through injecting drug use and in patients receiving PI-based or non-standard ART regimens. Our findings imply that early initiation of ART, which is now recommended for all PWH, could result in reduced occurrence of LLV.

## Acknowledgments

This study benefitted from data provided by the national quality register InfCareHIV.

## Author Contributions

**Conceptualization:** Per Björkman, Olof Elvstam.

**Data curation:** Olof Elvstam.

**Formal analysis:** Hanna Brattgård, Olof Elvstam.

**Resources:** Per Björkman, Piotr Nowak, Carl Johan Treutiger, Magnus Gisslén.

**Supervision:** Per Björkman, Olof Elvstam.

**Writing – original draft:** Hanna Brattgård.

**Writing – review & editing:** Hanna Brattgård, Per Björkman, Piotr Nowak, Carl Johan Treutiger, Magnus Gisslén, Olof Elvstam.

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
