## [Decision Letter · Decision Letter 0]

22 Mar 2022

PONE-D-22-03146Factors associated with low-level viraemia in people with HIV starting antiretroviral therapy: a Swedish observational studyPLOS ONE

Dear Dr. Elvstam,

Thank you for submitting your manuscript to PLOS ONE. After careful consideration, we feel that it has merit but does not fully meet PLOS ONE’s publication criteria as it currently stands. Therefore, we invite you to submit a revised version of the manuscript that addresses the points raised during the review process. This is an interesting study on an important and highly clinically relevant topic. I think this is excellent use of the data generated by your very good healthcare system.

The expert Reviewers raise some important questions we would be grateful if you could address in detail. Please give some thought to the figures and graphs, and clarity on definitions. Please also consider how the practice of HIV may have changed during your study period, impacting issues such as adherence.

We look forward to receiving your detailed response letter and a revised manuscript.

We look forward to receiving your revised manuscript.

Kind regards,

Jonathan Michael Schapiro

Academic Editor

PLOS ONE

Journal Requirements:

"I have read the journal's policy and the authors of this manuscript have the following competing interests: PB has received grants from Swedish State, grants from Region Skåne, grants from Gilead Nordic Fellowship during the conduct of the study, and personal fees from Gilead, outside the submitted work. CT has received honoraria as speaker and/or advisor from GlaxoSmithKline/ViiV, outside the submitted work. MG has received research grants from Gilead Sciences and Janssen-Cilag and honoraria as speaker and/or scientific advisor from Amgen, Biogen, Bristol-Myers Squibb, Gilead Sciences, GlaxoSmithKline/ViiV, Janssen-Cilag, MSD, Novocure, Novo Nordic and Sanofi, outside the submitted work. OE has received grant from Pfizer, outside the submitted work."

We note that you received funding from a commercial source: Swedish State, Region Skåne, Gilead Nordic Fellowship, Gilead Sciences, Janssen-Cilag and Pfizer

Reviewers' comments:

Reviewer's Responses to Questions

**Comments to the Author**

1. Is the manuscript technically sound, and do the data support the conclusions?

Reviewer #1: Partly

Reviewer #2: Yes

2. Has the statistical analysis been performed appropriately and rigorously? 

Reviewer #1: I Don't Know

Reviewer #2: Yes

3. Have the authors made all data underlying the findings in their manuscript fully available?

Reviewer #1: Yes

Reviewer #2: Yes

4. Is the manuscript presented in an intelligible fashion and written in standard English?

Reviewer #1: Yes

Reviewer #2: Yes

5. Review Comments to the Author

Reviewer #1: 1. justify the inclusion dates of 2006-2017

2. did you only include person who received 3 drug regimen

3. it is not clear how was LLV defined- did this occur any time during the 24 months after ART initiation or at 24 months- how did you deal with persons whose viral load was still suppressing but not yet to that level

4. I have concerns about the persons that were excluded because of viral failure or treatment interruption- might these persons have had LLV and then failed- could there be misclassification issues

5. the analysis was done by drug class- not sure how much adherence was playing a role and the early signal for PI related to adverse events of the earlier agents- also did you take into consideration the number of ARV drugs- was there any signal once the single tablet regimens became standard

6. in the same vein did the incidence of LLV decrease with calendar year with newer and more tolerable ART

7. in table 4 you have suggested that LLV is only 25% as common after 2014 and yet in the discussion state that LLV common in those with "modern" ART regimens

Reviewer #2: This is a well-written manuscript describing a well-designed retrospective study addressing the important topic of low-level viremia. The fact that the analysis is drawn from all adult patients in Sweden with HIV is a strength of the manuscript.

Comments:

1. Introduction: Par 1. "LLV confers increased risk of VF and has also been associated with all-cause mortality in a recent analysis of the nationwide Swedish cohort (5-10)." The references supporting the first part of the sentence should be separated from those supporting the second part of the sentence. The main finding of reference 10 should be described. Is the increase mortality a result of the LLV or a result of eventual VF?

2. The rationale for Tables 3 and 4 should be provided. Otherwise, their inclusion is confusing.

3. The authors don't address whether there is anyway to discern whether patients with LLV may have viremia resulting from clonal proliferation. For example is there a population of persons that have experienced prolonged VS only to have LLV then become detectable. This might suggest clonal proliferation of infected cells rather than viruses completing their replication cycle.

6. PLOS authors have the option to publish the peer review history of their article (what does this mean?). If published, this will include your full peer review and any attached files.

Reviewer #1: No

Reviewer #2: No

---

## [Author Response · Author response to Decision Letter 0]

27 Apr 2022

Comments of reviewer #1

• Comment 1: justify the inclusion dates of 2006-2017

Our response: Thank you for this comment. This manuscript is based on an excerpt from the InfCareHIV register restricted to persons who started ART between 2006 and 2017. More sensitive viral load assays (with a limit of quantification of ≤40 copies/ml) were increasingly in use in Sweden that year. We agree that it is important to consider the changing management of people with HIV over the study years when interpreting our results; however, we believe that this is addressed in the analysis of participants starting ART 2014 or later (the year when early ART was recommended for all people with HIV in Sweden, and when dolutegravir became available). Since we followed participants 30 months after start of ART, we were not able to include those who started ART 2018 or later. 

• Comment 2: did you only include person who received 3 drug regimen

Our response: We agree that this was not clear in the previous version. We now added (line 75-76): “(≥3 non-booster drugs, of which not all were nucleoside/nucleotide reverse transcriptase inhibitors)”. 

• Comment 3: it is not clear how was LLV defined- did this occur any time during the 24 months after ART initiation or at 24 months- how did you deal with persons whose viral load was still suppressing but not yet to that level

Our response: LLV could occur at any time during the observation period (6-30 months after start of ART). All participants who had ≥2 consecutive VL measurements of 50-199 copies/ml during the observation period were counted as LLV. Participants with virologic failure (≥2 consecutive VL measurements ≥200 copies/ml or any VL ≥1000 copies/ml) during the 24 months observation period were excluded, and those with isolated VL measurements of 50-999 copies/ml preceded and followed by VL<50 copies/ml were included in the suppression category.

We would like to clarify one situation which was not specified in the previous version: the classification of persons with 1 VL in the range 50-199 and 1 VL in the range 200-999 (at least 4 weeks apart). These were classified as LLV (in the same vein that a single VL <1000 copies/ml was allowed for viral suppression) if they did not meet the definition of failure during the studied period. This is now clarified (line 88-89): “(or one VL in the range 50-199 and one in the range 50-999 copies/ml, not meeting the definition of virologic failure)”.

• Comment 4: I have concerns about the persons that were excluded because of viral failure or treatment interruption- might these persons have had LLV and then failed- could there be misclassification issues

Our response: Thank you for this relevant comment; these data have now been added to the manuscript (line 120-122): “Of 251 individuals excluded since they met the definition of virologic failure during the observation period, 16 (6.4%) had LLV before reaching failure.”

For this study, we follow our participants for a limited time (30 months from ART initiation), and we argue that those who develop failure within this time frame likely represent a different entity than people with LLV without failure. Still, we agree that this is relevant to discuss further, and we added a section on this to the discussion (line 258-260): “Also, some individuals with LLV were excluded since they developed failure; our results are therefore restricted to people with LLV without treatment failure during the first years after ART initiation”.

• Comment 5: the analysis was done by drug class- not sure how much adherence was playing a role and the early signal for PI related to adverse events of the earlier agents- also did you take into consideration the number of ARV drugs- was there any signal once the single tablet regimens became standard

Our response: We agree that adherence could be an important factor, but unfortunately our material does not include any adherence data, as discussed in the manuscript (line 254-255). The early unboosted PI-based regimens with multiple doses per day and more side effects were rarely used in Sweden during the studied period (data not shown). We do not have data on coformulated tablets; thus, we cannot distinguish between single tablet regimens and other regimens. Still, it remains controversial whether single table regimens improve adherence, and whether a potential difference is clinically relevant (1). As a result, and since we did not want to include too many variables, we did not consider single table regimens in our analysis.

• Comment 6: in the same vein did the incidence of LLV decrease with calendar year with newer and more tolerable ART

Our response: Thank you for this suggestion. Following this comment, we calculated the proportion of LLV among people starting ART each calendar year. Although there is some variation, our data do not suggest a decreasing incidence of LLV during the study period, and it is possible that other factors (such as time between HIV acquisition and ART initiation and changing demographics of the Swedish HIV cohort) have a greater impact than the use of modern ART regimens. Since we found these results interesting, we decided to include them in the revised version: 

o Line 116: “Figure 2 was made using R (18) with the ggplot2 package (19)”

o Line 123-124: “Approximately similar proportions of LLV were observed across calendar year of ART initiation (Fig 2).” 

o Line 133-135: “Fig 2. Proportion of study participants experiencing LLV during 6-30 months after start of ART, grouped by year of ART initiation. Abbreviations: ART, antiretroviral therapy; LLV, low-level viraemia.”

o Figure 2

• Comment 7: in table 4 you have suggested that LLV is only 25% as common after 2014 and yet in the discussion state that LLV common in those with "modern" ART regimens

Our response: We now describe the frequency of LLV across calendar year in Figure 2. Table 4 is a stratified analysis where participants starting ART before/after 2014 are analyzed separately.

Comments of reviewer #2

• This is a well-written manuscript describing a well-designed retrospective study addressing the important topic of low-level viremia. The fact that the analysis is drawn from all adult patients in Sweden with HIV is a strength of the manuscript.

Our response: Thank you for your encouraging comments.

• Comment 1: Introduction: Par 1. "LLV confers increased risk of VF and has also been associated with all-cause mortality in a recent analysis of the nationwide Swedish cohort (5-10)." The references supporting the first part of the sentence should be separated from those supporting the second part of the sentence. The main finding of reference 10 should be described. Is the increase mortality a result of the LLV or a result of eventual VF?

Our response: Thank you for this suggestion; we have changed the references accordingly (line 55-57). The main finding of reference 10 is increased all-cause mortality for people with LLV (both LLV of 50-999 and 50-199 copies/ml). Of note, this is likely not a result of virologic failure, since study participants who developed failure were reclassified as high-level viraemia in that study. Having high-level viraemia (≥1000 copies/ml) was also associated with increased mortality in that study (2), as expected.

• Comment 2: The rationale for Tables 3 and 4 should be provided. Otherwise, their inclusion is confusing.

Our response: Women and men with HIV have certain demographic differences, and there are comparatively few data published on women with HIV, both regarding LLV and other areas. Therefore, we consider the analysis stratified by sex to be a strength of this study. The rationale behind Table 4 is that the clinical management of people with HIV has changed considerably during the studied period. 2014 was the year when dolutegravir was introduced in Sweden and also the year when ART was recommended irrespective of CD4 counts; accordingly, it is possible that contributing factors for LLV could be different before/after 2014. We have tried to explain this further (line 106-110): ”To investigate if the same factors were associated with LLV in men and women with HIV, respectively, we separately performed an analysis stratified by sex. Likewise, since the management of HIV has changed considerably during the studied years, we performed an analysis stratified by year of treatment initiation (before/after January 1, 2014, the year when dolutegravir became available in Sweden).”

• Comment 3: The authors don't address whether there is anyway to discern whether patients with LLV may have viremia resulting from clonal proliferation. For example is there a population of persons that have experienced prolonged VS only to have LLV then become detectable. This might suggest clonal proliferation of infected cells rather than viruses completing their replication cycle.

Our comment: Thank you for this interesting comment. To our knowledge, there is no way to distinguish between so called “monotypic” (e.g. related to clonal proliferation) and “diverse” (with signs of ongoing replication) LLV without analyzing viral sequences (or perhaps 2-LTR circles as a marker of replication). Of note, we studied LLV during the first 30 months after ART initiation. Consequently, our study is not designed to study individuals with longer periods of viral suppression before LLV. 

 

References

1. Nachega JB, Parienti JJ, Uthman OA, Gross R, Dowdy DW, Sax PE, et al. Lower pill burden and once-daily antiretroviral treatment regimens for HIV infection: A meta-analysis of randomized controlled trials. Clin Infect Dis. 2014;58(9):1297-307.

2. Elvstam O, Marrone G, Medstrand P, Treutiger CJ, Sonnerborg A, Gisslen M, et al. All-Cause Mortality and Serious Non-AIDS Events in Adults With Low-level Human Immunodeficiency Virus Viremia During Combination Antiretroviral Therapy: Results From a Swedish Nationwide Observational Study. Clin Infect Dis. 2021;72(12):2079-86.

---

## [Editor Report · Decision Letter 1]

3 May 2022

Factors associated with low-level viraemia in people with HIV starting antiretroviral therapy: a Swedish observational study

PONE-D-22-03146R1

Dear Dr. Olof Elvstam,

You and your co-authors have addressed the reviewer comments very appropriately and have provided a very high level and important manuscript.

We’re pleased to inform you that your manuscript has now been judged scientifically suitable for publication and will be formally accepted for publication once it meets all outstanding technical requirements.

Kind regards,

Jonathan M Schapiro

Academic Editor

PLOS ONE

---

## [Editor Report · Acceptance letter]

9 May 2022

PONE-D-22-03146R1 

Factors associated with low-level viraemia in people with HIV starting antiretroviral therapy: a Swedish observational study 

Dear Dr. Elvstam:

I'm pleased to inform you that your manuscript has been deemed suitable for publication in PLOS ONE. Congratulations! Your manuscript is now with our production department. 

Kind regards, 

on behalf of

Dr. Jonathan Michael Schapiro 

Academic Editor

PLOS ONE